# Detection of Viral −RNA and +RNA Strands in Enterovirus-Infected Cells and Tissues

**DOI:** 10.3390/microorganisms8121928

**Published:** 2020-12-04

**Authors:** Sami Salmikangas, Jutta E. Laiho, Kerttu Kalander, Mira Laajala, Anni Honkimaa, Iryna Shanina, Sami Oikarinen, Marc S. Horwitz, Heikki Hyöty, Varpu Marjomäki

**Affiliations:** 1Department of Biological and Environmental Science/Nanoscience Center, University of Jyväskylä, Survontie 9C, FI-40500 Jyväskylä, Finland; sami.salmikangas@helsinki.fi (S.S.); kerttu.o.kalander@student.jyu.fi (K.K.); mira.a.laajala@jyu.fi (M.L.); 2Faculty of Medicine and Health Technology, Tampere University, FI-33520 Tampere, Finland; jutta.laiho@tuni.fi (J.E.L.); anni.honkimaa@tuni.fi (A.H.); sami.oikarinen@tuni.fi (S.O.); heikki.hyoty@tuni.fi (H.H.); 3Department of Microbiology and Immunology, Life Sciences Institute, University of British Columbia, Vancouver, BC V6T1Z3, Canada; ishanina@mail.ubc.ca (I.S.); mhorwitz@mail.ubc.ca (M.S.H.)

**Keywords:** antiviral drugs, branched DNA, enterovirus, in situ hybridization, negative RNA, positive RNA, replication

## Abstract

The current methods to study the distribution and dynamics of viral RNA molecules inside infected cells are not ideal, as electron microscopy and immunohistochemistry can only detect mature virions, and quantitative real-time PCR does not reveal localized distribution of RNAs. We demonstrated here the branched DNA in situ hybridization (bDNA ISH) technology to study both the amount and location of the emerging −RNA and +RNA during acute and persistent enterovirus infections. According to our results, the replication of the viral RNA started 2–3 h after infection and the translation shortly after at 3–4 h post-infection. The replication hotspots with newly emerging −RNA were located quite centrally in the cell, while the +RNA production and most likely virion assembly took place in the periphery of the cell. We also discovered that the pace of replication of −RNA and +RNA strands was almost identical, and −RNA was absent during antiviral treatments. ViewRNA ISH with our custom probes also showed a good signal during acute and persistent enterovirus infections in cell and mouse models. Considering these results, along with the established bDNA FISH protocol modified by us, the effects of antiviral drugs and the emergence of enterovirus RNAs in general can be studied more effectively.

## 1. Introduction

The genus *Enterovirus* belongs to the family of *Picornaviridae*, and they are classified into 15 species [1]. Over 280 different enterovirus types can infect humans, and they all belong to the species Enterovirus A-D (≈115 types) or Rhinovirus A-C (≈170 types) according to their molecular and/or antigenic characteristics. Enteroviruses include, for example, coxsackieviruses, polioviruses, rhinoviruses, and echoviruses, and they can cause a range of diseases from minor common colds and rashes to severe conditions such as myocarditis, meningitis, and encephalitis [2]. Some chronic diseases, such as atherosclerosis and type 1 diabetes, have also been linked to enteroviruses [3,4]. Despite the prevalence of diseases caused by enteroviruses, enterovirus vaccines have thus far only been developed against poliovirus and enterovirus 71. Some antiviral drugs have been or are being developed against enteroviruses, but none have made it to commercial markets yet.

Enteroviruses are small (≈30 nm), non-enveloped RNA viruses that have high mutation and recombination rates [5]. They have icosahedral capsids consisting of 60 repeating protomers, which are made from the viral capsid proteins VP1, VP2, VP3, and VP4. Inside the capsid, enteroviruses have a single-stranded positive-sense RNA genome (+RNA). The life cycle of enteroviruses starts with attachment, where the virus binds to the cell surface receptor(s) (reviewed more in depth in [1]). The binding causes receptor-mediated endocytosis of the virus particle, after which a change in pH (endosome maturing), receptor binding, or ionic changes [6] will lead to uncoating of the virus. During uncoating, the viral genome is first released from the capsid into the endosome, and then transported through the endosomal membrane into the cytosol. There, the +RNA is translated, resulting in a single polyprotein, which is subsequentially cleaved by viral proteases 2A^pro^, 3C^pro^, and 3CD^pro^ into capsid proteins VP0 (intermediate polyprotein for VP2 and VP4), VP1, and VP3 and replication proteins 2A-2C and 3A-3D. The viral genome replication is started by the RNA polymerase 3D^pol^ protein, which synthesizes a negative RNA strand (−RNA) to serve as a template for new +RNA strands. The −RNA and +RNA strands form both stable and unstable double-stranded RNA (dsRNA) intermediates as well. The newly synthesized +RNA strands are then used to generate new −RNA strands, translated to create more capsid proteins, and/or assembled into new provirions with the capsid protein pentamers. The pentamers are created by combining five capsid protomers, which include the VP0, VP1, and VP3 capsid proteins, into a pentamer.

After assembling the viral +RNA and translated and cleaved capsid proteins VP0, VP1, and VP3 into a provirion, the RNA induces the cleavage of VP0 into VP2 and VP4, resulting in a mature and infective virus particle [7]. Some of the pathways and mechanisms underlying the way in which the virus induces cell lysis are still unclear. What we do know is that viral proteases 2A^pro^ and 3C^pro^ induce host shut-off, interrupt interferon and stress pathways, and disrupt the cytoskeleton and nucleocytoplasmic transport, ultimately leading to cell lysis [8]. Newly synthesized, mature, and infective viruses are released during the cell lysis into the surrounding extra-cellular matrix and make their way to infect new cells. In recent studies, it has been shown that some enteroviruses are also capable of exiting the cell in extracellular vesicles in a non-lytic fashion [9]. The genome replication and translation are highly conserved between different enterovirus species, making these steps ideal targets for antiviral drugs [1].

The current methods and technology to study the distribution and dynamics of viral RNA molecules inside an infected cell are not ideal, as electron microscopy and immunohistochemistry can only detect mature virions, and quantitative real-time PCR (qPCR) is not suited to study the localized distribution and dynamics of RNAs. In situ hybridization (ISH) techniques try to remedy this situation but are limited by low signal intensity and nonspecific probe binding. Branched DNA ISH technology (bDNA ISH) [10], applied for example in RNAScope (ACDBio) and ViewRNA (Thermo Scientific, previously by Affymetrix [11]) techniques, has a higher signal intensity and more specific probe binding than regular ISH. The bDNA ISH technology relies on hybridizing tiers of DNA oligos on top of a target RNA probe [10]. The subsequent preamplifier and amplifier DNA oligos create a “branched tree” type of DNA structure, where up to 8000 fluorophore-conjugated label probes can then be hybridized to. This is sufficient to detect even single RNA molecules with a confocal microscope, whereas traditional, non-amplified fluorescent ISH (FISH) techniques require a 600 times longer exposure and a 100 times greater camera gain than bDNA FISH to be able to see similar discernible spots. This technique, when using a specific kit, can also be coupled with standard antibody-based immunofluorescence to detect, for example, viral capsid proteins.

Along with vaccines, antiviral drugs have shown great promise in the treatment of diseases caused by viruses. As of yet, there are no vaccines or antiviral drugs against non-polio enteroviruses on the market, with the exception of an inactivated Enterovirus 71 vaccine licensed in China [12]. In addition, group B coxsackieviruses (CVBs) are important vaccine targets, and clinical trials with multivalent CVB vaccine start in 2020 [13]. In addition to studying the distribution and amounts of enterovirus RNAs, we decided to test the effect of two different drugs on the amount and distribution of enterovirus RNAs during an infection in vitro. The drugs used were Rac1 inhibitor NSC23766 (Rac1-I) and calpain inhibitor 1 (Cal-I1), and they have previously been shown to have antiviral activity and prevent the virus from starting the replication [14,15]. These antivirals target cellular host factors instead of the virus particles themselves, creating a higher barrier for the virus to develop drug resistances [16].

Rac1 is a small cellular Rho GTPase protein that regulates the amount and distribution of guanosine triphosphate (GTP), an important cellular signaling molecule, by hydrolyzing it to guanosine diphosphate (GDP) [17]. By modifying the actin cytoskeleton, GTPases regulate many cellular processes, such as motility, gene expression, and cell cycle, but have also important supportive and suppressive functions in viral life cycles [18]. It has been shown that Rac1 is involved in the regulation of the entry of enteroviruses into the cell [19], and thus is an important host factor in enterovirus infections. The Rac1 inhibitor NSC23766 specifically inhibits Rac1 without effecting other proteins of the Rho GTPase family [20], making it an ideal candidate for an antiviral drug.

Calpains are calcium-dependent cysteine proteases expressed in all mammals [21]. Their normal role in the cell is to catalyze the processing of cytoplasmic proteins, but they have a role in the replication cycle of enteroviruses as well [14,22]. Unlike the Rac1 protein described above, calpains 1 and 2 are not required for the entry of the virus, but rather at later stages of the infection in RNA replication [14]. It was recently demonstrated that cellular calpains can cleave viral capsid proteins VP1 and VP3 from the enterovirus polyprotein [23], and that enterovirus infection upregulates the activity of calpains, suggesting that calpains are essential for the cleavage of the viral polyprotein. Laajala et al. [23] also demonstrated high cross-reactivity of calpain inhibitor 1 with the viral proteases 2Apro and 3Cpro, making it a promising antiviral drug.

We have previously applied the bDNA ISH technology to detect enterovirus +RNA in infected cells and mouse pancreata, with specifically designed probes [24], and have also tested the relative sensitivity of this technique to detect enteroviruses (+RNA) in formalin-fixed, paraffin-embedded samples (FFPE) [25]. The bDNA FISH technology has also been successfully used to detect both positive and negative sense viral RNAs [26,27]. Since enteroviruses generate only small amounts of −RNA, about 40- to 100-fold less than +RNA [28], and because the RNA strands form double stranded RNA intermediates, the −RNA can be difficult to detect, even with bDNA FISH techniques.

In this study we applied the previously established bDNA ISH for enterovirus-infected paraffin samples [24] to detect +RNA and −RNA in acutely and persistently infected cells and mice pancreata to evaluate the functionality of −RNA probe and to see the relative amounts of the two strands during the infection. We also set out to establish a protocol to detect both −RNA and +RNA molecules during an early enterovirus infection in vitro using bDNA FISH and confocal microscopy, and to study the cellular distribution and dynamics of the mentioned RNA molecules during an enterovirus infection. We further used reverse transcription quantitative real-time polymerase chain reaction (RT-qPCR) in addition to bDNA FISH in order to determine the relative amounts of enterovirus RNAs during an infection in vitro.

The distribution, dynamics, and localization of enterovirus RNAs during an infection (both acute and persistent) are quite unknown to this date, mostly due to the lack of techniques to study them. This is particularly true for the rare −RNA strands, since the amount of −RNA during the course of an enterovirus infection is very limited, making it even harder to detect. Studying the distribution, dynamics, and amounts of enterovirus RNAs (both + and -) during an infection with bDNA ISH techniques and RT-qPCR can provide valuable information about the infection itself and about how to target the viral life cycle with antiviral drugs, and moreover can help to develop novel methods to study enterovirus RNAs and infections.

## 2. Materials and Methods

### 2.1. Cell Culturing and Virus Infections in Cells

Human alveolar basal epithelial A549 cells (ATCC) were used in the bDNA FISH experiments and cultivated at 37 °C in Dulbecco’s modified Eagle’s medium (DMEM, Gibco Life Technologies, Waltham, MA, USA) ref. 52100-039) supplemented with 10% fetal bovine serum (FBS, Gibco Life Technologies, Ref. 10270-106), 1% L-glutamine (Gibco Life Technologies, ref. 35050-038), and 1% penicillin and streptomycin (Gibco Life Technologies, ref. 15130-122).

Coverslips, 70–80% confluent with cells, were transferred to sterile 4-well plates (Thermo Scientific, Waltham, MA, USA ref. 176740) and infected with in-house purified coxsackievirus A9 (CVA9) (Griggs strain, infectivity 1.6 × 10^11^) by diluting the virus 1:3000 in DMEM supplemented with 1% FBS (with 1.07 × 10^7^ plague forming units (PFU) on the coverslip). Viruses were allowed to bind to the cells for 45 min on ice, following a wash with cold 0.5% bovine serum albumin (BSA) in phosphate-buffered saline (PBS) before starting the infection to ensure an even infection rate. The infection was started by changing the media to DMEM supplemented with 10% FBS and incubating the cells at 37 °C for 1–6 h, depending on the experiment. The infected cells were then fixed with the ViewRNA Cell Plus Assay kit’s (Invitrogen, Carlsbad, CA, USA) ref. 88-19000-99) Fixation/Permeabilization Solution for 30 min at room temperature (RT).

Persistent infection in pancreatic ductal (PANC-1) cells was established as previously described [29]. Briefly, PANC-1 cells were initially infected with a wild type CVB1 (strain 10796) using a low titer of the virus, leading to a strong cytopathic effect (CPE) and extensive cell death. Remaining living cells were transferred on fresh PANC-1 cells. A strong CPE was again seen after cells were added on fresh cells, after which the remaining living cells were maintained with regular washes and passaged once a week by using cell scraper until the persistence became established. For ISH and immunofluorescence (IF) analyses, the cells were harvested after 172 days of viral persistence by scraping and fixed in 10% formalin for 24 h prior to dehydration and paraffin embedding. A similar sample was collected 24 h after the initial infection, representing the acute stage of the infection.

### 2.2. Virus Infected Mice

Virus stocks of CVB1 (strain 10796) were prepared on monolayers of HeLa cells using a multiplicity of infection of 0.1 in DMEM. Virus was collected following freeze–thaw cycles, filtered, and stored at −80. Viral titers were measured using standard plaques assay on HeLa cell monolayers. Mda5/NOD (het) mice were infected internationally with sublethal dose of 10^5^ pfu CVB1-10796 per mouse and harvested on day 3 post-infection. Six mice were infected, and non-infected Mda5/NOD (het) mice were used as controls.

### 2.3. Immunofluoresence

IF for paraffin samples was performed as previously described [30] by using an in-house polyclonal antibody produced in rabbit against viral capsid protein VP3 of CVB4 Tuscany strain [25] and commercial goat anti-rabbit IgG (H+L) Alexa Fluor 568 secondary antibody (Thermo Scientific). IF for bDNA FISH coverslips was performed with the ViewRNA Cell Plus Assay kit according to the manufacturer’s instructions, using either commercial or in-house primary antibodies (CVA9-8863, CVA9-861, CVA9-K3, and J-2) and commercial fluorescent secondary antibodies (goat anti-rabbit IgG Alexa Fluor 488, goat anti-mouse IgG Alexa Fluor 488, and goat anti-mouse IgG Alexa Fluor 633; antibodies used are detailed in Appendix A). Notably, CVA9-861 and CVA9-8864 primary antibodies (kind gifts from Dr. Merja Roivainen, Finnish Institute for Health and Welfare, Helsinki, Finland) were used to label the viral capsid protein VP1, and the J2 primary antibody (Scicons, Hungary) was used to label the viral dsRNA.

### 2.4. In Situ Hybridization

bDNA FISH for coverslips was performed with the ViewRNA Cell Plus Assay Kit by Thermo Scientific according to the manufacturer’s instructions, with the exception of a 15 min incubation with 95% formamide in 0.1% SSC buffer at 65 °C before the addition of the primary RNA probes. For paraffin samples, we used ViewRNA Tissue kit (Thermo Scientific) according to the manufacturer’s instructions and as previously described [24], with the exception of tissue boiling and protease incubation times for mouse pancreata, which were 10 and 15 min, respectively. Previously published in-house/custom designed probe sets to detect +RNA (named EVAB+) were used [25]. For the present study, we produced a complementary probe to the +RNA to detect −RNA (named EVAB−; EVAB+ and EVAB− probe set sequences are detailed in Appendix A). A detailed protocol for performing the bDNA FISH experiment is provided in Appendix B.

### 2.5. Microscopy

Before imaging the cells with the confocal microscope, we stained the nuclei of the cells with 4′,6-diamidino-2-phenylindole (DAPI) (Invitrogen/Molecular Probes, ref. D3571) diluted 1:40,000 in 1x PBS for 5 min at RT, mounted on microscopy slides (Thermo Scientific) with in-house-made Mowiol-Dabco (33.3% (*v/v*) glycerol containing 16.6% (*w/v*) Mowiol (Calbiochem, St. Louis, MO, USA) ref. 475904) and 2.5% (*w/v*) 1,4-diazabicyclo [2.2.2]octane (DABCO) (Sigma Aldrich, St. Louis, MO, USA) ref. D2, 780-2), and stored in the dark at 4 °C overnight.

The cells were imaged using an Olympus microscope IX81 with FluoView-1000 confocal setup, with the Olympus UPlanFLN 40x oil immersion objective (numerical aperture 1.30) and Olympus Immoil-F30CC immersion oil. The lasers used were a 405 nm multiline diode laser, a 488 nm argon laser, and 543 nm and 633 nm HeNe lasers. The following software settings were used in image acquisition: unidirectional scan mode, dwell time—4.0 µs/pixel, image size—640 × 640 pixels, aspect ratio—1:1, sequential line capture, Kalman averaging with 3 scans.

ISH-stained paraffin sections were imaged using an Olympus BX60 microscope fitted with an Olympus Colorview III camera.

### 2.6. Antiviral Drugs

For the RT-qPCR and bDNA FISH experiments with the antiviral drugs, the drugs were added to the 10% DMEM, which was administered to the cells right before the start of the infection. Rac1-I (Sigma Aldrich, Ref. SML0952) was used at 200 µM final concentration and Cal-I1 (Roche Diagnostics GmbH, Basel, Switzerland ref. 11086090001) was used at 200 µM final concentration. Cell viability assays to determine the cytotoxicity of the drugs was performed with Cell Titer-Glo according to the manufacturer’s instructions (Promega, Madison, WI, USA).

### 2.7. RT-qPCR

The RT-qPCR experiment was started with seeding 90,000 A549 cells per well on 4-well plates. After 24 h, the plates were infected with 1:10,000 diluted CVA9 virus (final pfu of 3.2 × 10^6^ on well) as described in Section 2.1 (400 µL final volume). After incubation at 37 °C for 0–5 h, the infection was stopped by placing the plates in a −80 °C freezer.

After stopping the infection, the plates were thawed for 15 min at 37 °C and frozen for 15 min at −80 °C. This was repeated three times to lyse the cells and release the virus without destroying the viral RNA. Cell debris (400 µL) was transferred to a 1.5 mL LoBind Eppendorf tube (Eppendorf, Hamburg, Germany, ref. 0030108051) and centrifuged for 10 min at 16,700× *g* with a table centrifuge (Eppendorf, Centrifuge 5415 D). A total of 140 µL of the supernatant was used for the RNA extraction, and the rest was stored at −80 °C.

Viral RNA was extracted with QIAamp Viral RNA Mini kit (Qiagen, Hilden, Germany ref. 52906) according to the spin protocol provided in the kit’s handbook. Extracted RNA (60 µL) was stored at −80 °C.

The primers for the RT reaction and the qPCR reaction were synthesized and acquired from Thermo Scientific. The primer for the −RNA had the following sequence: 5′-GAAACACGGACACCCAAAGTA-3′, and the primer for the +RNA had the following sequence: 5′-CGGCCCCTGAATGCGGCTAA-3′. The sequences for the primers were originally received from Matti Waris (University of Turku) and the sequences have been successfully used before to detect enterovirus RNAs.

The RT reaction was performed by making a master mix (Appendix A) according to the manufacturer’s instructions, adding 10 µL of the extracted RNA template on the 30 µL of the master mix to incubate for 1 h at 42 °C, following a heat-inactivation of the RT enzyme for 10 min at 70 °C. The acquired complementary DNA (cDNA) was diluted 1:100 in nuclease-free water and stored at −20 °C.

Control reactions for the RT-qPCR experiment included negative controls for RNA extraction, negative controls for the RT reaction, and negative controls for the qPCR reaction. In negative controls, we used nuclease-free water (Alfa Aesar, Haverhill, MA, USA, ref. J71786) instead of the template RNA/cDNA.

The qPCR was performed by making a master mix (Appendix A) according to the manufacturer’s instructions and adding 5 µL of the cDNA template on 20 µL of the master mix on PCR plates (Bio-Rad, Hercules, CA, USA, ref. HSL9601). Triplicate wells were made for all samples. Template and the master mix were thoroughly mixed in the wells, and the plate was carefully sealed with a PCR plate tape (Bio-Rad, ref. MSB1001). The qPCR was performed with a Bio-Rad CFX96 Touch Real-Time PCR Detection System with the following protocol: (1) 95 °C, 10 min; (2) 95 °C, 15 s; (3) 60 °C, 1 min; (4) repeat steps 2–3 for 39 more times; (5) 12 °C, 10 min.

### 2.8. Image and Data Processing

The images acquired from the confocal microscope were refined and analyzed with Fiji v. 1.52s, a distribution of the ImageJ image processing software [31]. First, all images were converted to 8-bit images to have comparable intensity values between 1 and 256. The minimum intensity value of uninfected negative control cell images was manually set so high that the image was completely blank, using a Hi-Lo lookup table. This minimum intensity threshold was then applied to all the sample images, effectively removing background fluorescence. In addition, a 1.0-pixel radius mean filter was applied to the images to remove unwanted background noise. Equal amounts of brightness were also applied to all images to make the signals clearer while keeping the amounts of signal comparable between images. Colocalization analyses were performed with Manders’ colocalization correlation algorithm in Fiji by using the Coloc 2 plugin after thresholding the background, as explained above.

Intensity calculations from bDNA FISH images were acquired by using the “Measure” tool with “Limit to threshold” in Fiji after thresholding the background as explained above. Approximately 100 cells per image was used for the calculations. The measurement calculates the area of all pixels that have intensity values above the set threshold, which is a quantitation of the amount of target transcripts in the sample image. According to the manufacturer, each dot in bDNA FISH images represents one target transcript, and the level of expression can be quantified by measuring the number of dots per cell, meaning that the amplification of the signal is a linear function. It is important to note that experimental conditions, such as the microscope and camera used, can affect the visibility of the RNAs, meaning that the results are comparable only with studies with similar settings.

For the RT-qPCR data, average Cq values were calculated from the triplicate measurements for each sample and plotted as a bar chart with standard deviation error bars representing the amount of variance in the Cq values in the triplicates.

## 3. Results

During the optimization of the bDNA FISH method, we discovered that the −RNA was barely visible under the confocal microscope, possibly due to the +RNA and −RNA strands making double-stranded RNA intermediates, effectively inhibiting the binding of our RNA probes to the strands. To overcome this problem, we tested 100% methanol, up to 8 molar urea, and 95% formamide in 0.1x SSC buffer to separate the strands from each other. The results (not shown here) indicated that 100% methanol and up to 8 molar urea did not affect the visibility of either RNA strand, but that 95% formamide in 0.1· SSC buffer improved the visibility of the −RNA significantly. Using high temperature combined with 95% formamide to separate the RNA strands was critical for the success of our bDNA FISH experiments.

None of the negative controls used for RT-qPCR crossed the Cq value threshold, proving that the samples were not contaminated by other RNA- or DNA molecules.

### 3.1. Distribution of Enterovirus +RNA and −RNA in Relation to the Capsid during an Infection

In our first experiments, we used standard antibody-based IF to label the CVA9 capsid protein VP1 and bDNA FISH to label either the +RNA or the −RNA at timepoints of 1, 2, 3, 4, and 5 h post-infection (p.i.). The results indicated that the amount of viral capsid protein was negligible before 4 h p.i., and most of the capsid protein was located peripherally in the cell (Figure 1 and Figure 2). The +RNA was visible even at 1 h p.i., but the amount of visible +RNA surged at 3 h p.i. and gradually rose at 4 h and 5 h p.i (Figure 1). These results suggest that the visible +RNA at 1–2 h p.i. originated from the input virus, which meant that the virus did not start replicating +RNA until after 2–3 h p.i. As seen from the timepoints 4–5 h p.i., the +RNA in the cell was located even more peripherally than the capsid protein, and around 30% of the capsid colocalized with the +RNA (Manders’ colocalization coefficients M1: 0.303, M2: 0.336). Similar results were achieved with coxsackievirus B3 (CVB3)- and echovirus 1 (EV1)- infected cells (not shown here).

In the samples where we labelled −RNA instead of +RNA, the amount of −RNA was negligible at timepoints 1–3 h p.i., but some −RNA could be seen after 4 h p.i. (Figure 2). As with the +RNA samples, the capsid became visible at 3–4 h p.i., and was located peripherally in the cell. The −RNA was located more closely to the nucleus than the capsid protein, and the capsid protein did not colocalize with the −RNA at all (Manders’ colocalization coefficients M1: 0.003, M2: 0.093). Similar results were achieved with CVB3- and EV1-infected cells (not shown here).

### 3.2. Distribution of Enterovirus +RNA, −RNA, and dsRNA during an Infection

Next, we simultaneously labelled cell nuclei and viral dsRNA with IF, and −RNA and +RNA with bDNA FISH from cells infected with CVA9 at timepoints 0.5, 1, 2, 3, 4, and 5 h p.i. (Figure 3). The results were similar to the first experiments, with −RNA becoming visible at 3–4 h p.i., input +RNA being visible at 0.5–2 h p.i., and the amount of +RNA surging at 3 h p.i. The amount of visible dsRNA was negligible at timepoints 0.5–2 h p.i., but surged in visibility at 3 h p.i.

According to our results, there was very little colocalization between the dsRNA and the −RNA (Manders’ colocalization coefficients M1: 0.013, M2: 0.043), even though they seemed to mostly reside in the same cellular locations (Figure 3). The majority (97%) of −RNA colocalized with the +RNA, but only around 2% of the +RNA colocalized with the −RNA (Manders’ colocalization coefficients M1: 0.973, M2: 0.019). Similarly, the majority (89%) of dsRNA colocalized with the +RNA, but only around 5% of the +RNA colocalized with the dsRNA (Manders’ colocalization coefficients M1: 0.892, M2: 0.048). These results indicate that the +RNA signal was quite ubiquitous, and that the colocalizations of both −RNA and dsRNA with the +RNA were marginal. The dsRNA and −RNA were more centrally located in the cell around the nucleus than the peripherally located +RNA. None of the RNA signals localized in the nuclei of the cells.

### 3.3. The Amounts of Enterovirus +RNA and −RNA during an Infection

We used RT-qPCR to detect the amounts of enterovirus RNA molecules in CVA9-infected cells at timepoints 0, 1, 2, 3, 4, and 5 h p.i. (Figure 4). The results are presented in Cq values, which represent the number of PCR cycles it takes for the RNA to cross a detectable threshold, meaning that the lower the value is, the more RNA there is in the sample. The results were in line with the bDNA FISH experiments, with the amount of −RNA being considerably lower than the amount of +RNA, and the amounts of both +RNA and −RNA increasing dramatically at 3–4 h p.i. A concentration series of RNA isolated from purified CVA9 and detected by qPCR is found in Appendix A. According to the concentration series, the amount of +RNA after 5 h (Figure 4) was in the range of 1 × 10^−4^ ng, while that of −RNA was in the range of 1 × 10^−6^ ng.

The amounts of enterovirus RNAs were also calculated from intensity values of the bDNA FISH sample pictures using Fiji. The results are presented in pixels that have intensity values above the set threshold plotted against time (Figure 5 and Figure 6). The results were quite similar to the results from our qPCR experiments, showing exponential growth in the amounts of both RNAs. These results confirm that there was approximately a 100-fold difference in the amounts of enterovirus −RNA and +RNA consistently throughout the infection.

### 3.4. Effect of Antiviral Drugs on the Amounts of Viral +RNA and −RNA

We tested the effect of the antiviral drugs Rac1-I and Cal-I1 on the amount of enterovirus +RNA and −RNA molecules being produced in CVA9-infected cells at 5 h p.i. (Figure 7). The results show that both of the drugs lowered the amount of viral +RNA and −RNA molecules to the levels of 0h infected cells (only input virus, no incubation period) and there was very little variation in the efficiency of the drugs.

The cytotoxicity of the drugs Rac1-I and Cal-I1 was determined with a cell viability assay. After a 5 h incubation with the drugs, only a 10% drop in cell viability was detected with Cal-I1 and no noticeable drop in cell viability was detected with Rac1-I in comparison with a DMSO control (Appendix A). This proves that rather than having cytotoxic effects, the drugs act directly as antivirals.

### 3.5. Effect of Antiviral Drugs on the Distribution of Viral +RNA and −RNA

We infected cells with CVA9 while the cells were under the effects of either Rac1-I or Cal-I1, and then simultaneously labelled the viral dsRNA, +RNA, and −RNA at timepoints 0.5 h p.i. and 5 h p.i. (Figure 8). The results indicate that Rac1-I completely blocked the virus infection in most cells in comparison to the control infection (Figure 8, panels A and C), but some cells were still infected, suggesting that some cells were leaky with the higher amount of viruses used in the microscope experiment (Figure 8, panel A). In the infected cells, the levels of all RNA molecules dropped considerably, but the localization and colocalization of the RNA molecules seemed to stay the same.

Cal-I1 seemed to be even more effective than Rac1-I, blocking the virus infection completely in almost all cells (Figure 8, panel B). Some cells showed input virus levels of +RNA, but no dsRNA or −RNA at all.

### 3.6. The Pace of Viral +RNA and −RNA Synthesis during a CVA9 Infection

Assuming that the efficiency of our qPCR was close to 100%, the amount of cDNA (from the viral RNA) was doubled every PCR cycle. This can be presented with the equation
2^x^ = y,(1)
where x is the number of PCR cycles and y is fold change. This can be further solved to
x = log2 y,(2)
which solves further in to
x = log y/log 2.(3)

The amounts of −RNA and +RNA compared to the timepoint 0 h p.i. (background level) were calculated with Equation (3) from the Cq data in Figure 4 and plotted against time 0–5 h p.i. (Figure 9). The fold changes of both RNAs during early infection (1–3 h p.i.) were negligible. At 4 h p.i., the amount of +RNA was 3.6 times higher (compared to 0 h p.i.) and the amount of −RNA was 2.4 times higher, and at 5 h p.i., the amount of +RNA was 13.4 times higher and the amount of −RNA was 12.4 times higher.

### 3.7. Viral +RNA and −RNA Detection in Formalin-Fixed, Paraffin-Embedded Samples of CVB1-Infected Cell Cultures and CVB1-Infected Mouse Pancreas

We applied ViewRNA ISH for formalin-fixed, paraffin-embedded (FFPE) samples to detect viral +RNA and −RNA in CVB1-infected and non-infected cells and mouse pancreas. In the first phase, we showed that the method detected both RNA strands in acutely and persistently CVB1-infected FFPE PANC-1 cells (Figure 10). Both +RNA and −RNA were clearly visible in either infection models, and +RNA was more strongly expressed than −RNA. In acutely infected cell culture, the +RNA was detected virtually in every cell. In persistently infected cell culture, strong punctual +RNA-positive cells were observed (about 10% of cells showed this pattern), surrounded by more weakly positive cells. Many negative cells were also present. Non-infected cells were always negative (Figure 10).

Next, FFPE pancreas tissue samples from CVB1-infected and mock-infected mice were analyzed. A strong +RNA expression was seen throughout the exocrine part of the pancreas. The −RNA expression was also clear, but less strong compared to +RNA expression. Pancreas from mock-infected mice was negative for both + and −RNA (Figure 10).

## 4. Discussion

Enteroviruses, a genus of the *Picornaviridae* family, are small, non-enveloped viruses with a single-stranded positive-sense RNA genome. While their life cycle is relatively well-known, the distribution and the amounts of the viral +RNA and −RNA strands during an infection has been clouded due to insufficient methods to study them. Here, we have shown that the amounts of both +RNA and −RNA are negligible before 3–4 h p.i. (Figure 3 and Figure 4) using ViewRNA FISH and RT-qPCR, meaning that the entry and uncoating of the virus particle and the initial translation of the input +RNA strand takes around 3–4 h. After this, the replication process starts and the amounts of both +RNA and −RNA surge dramatically at 4 h p.i. and 5 h p.i. These results are in line with previous studies stating that the majority of uncoating of CVA9 happens around 2 h p.i., while the replication cycle starts around 3 h p.i. [15]. The FFPE samples of acutely CVB1-infected samples were collected at a later stage of the infection (≈24 h p.i. for the cells and 3 days p.i for the mice) when the virus had already spread to most of the cells. This was clearly reflected by the frequent detection of both +RNA and −RNA in these cells. Both +RNA and −RNA were expressed in the entire pancreas of CVB1-infected mice. The amount of +RNA was notably higher compared to −RNA in both cells and mice.

In persistently infected samples, CVB1 +RNA and −RNA were clearly detected as individual highly positive cells, and the results were representative of the previously published observations of carrier-type persistence with high titers of virus produced but only in a small proportion of cells infected [32,33]. The amount of +RNA was higher compared to −RNA, which has also been observed in another model of carrier-state persistent infection [34].

From RT-qPCR results, we measured a 5–6 cycle difference in −RNA and +RNA amounts at all timepoints of acute infection (Figure 4), which, when calculated with Equation (3), translates to around a 32-fold to 64-fold difference between the amounts of −RNA and +RNA in fluorescence acute infection study on cover slips. However, since RT-qPCR does not give absolute amounts of measured RNAs, these results are only rough estimations. To further estimate the difference in the amounts of enterovirus −RNA and +RNA throughout the infection, we calculated the intensity values of the RNAs from the bDNA FISH images (Figure 5 and Figure 6). The results showed that there was a 100-fold difference in the amounts of enterovirus −RNA and +RNA consistently throughout the infection, confirming previous estimations of a 40- to a 100-fold difference [28].

We expected to see an increase in the amount of −RNA before the increase of +RNA, and that the pace at which the −RNA was replicated would drop off once the replication of +RNA started, but that seems to not have been the case (Figure 9). Interestingly, our results showed that the amounts of +RNA and −RNA increased at roughly the same pace, contradictory to earlier beliefs of +RNA synthesis being of a faster pace than −RNA synthesis [35]. Moreover, the replication pace of either of the RNAs did not slow down at any point during the infection, even though the −RNA was not used for anything after the replication process was complete. This may reflect, however, the presence of a stable −RNA in the dsRNA intermediate for longer times.

Since enterovirus virions have no inherent −RNA strands, the RT-qPCR studies should give no Cq value at all for the −RNA at 0 h p.i. However, we did obtain values for the −RNA at 0 h p.i. in our study (Figure 4 and Figure 7), even though we did not obtain values for any negative control reactions, indicating that false-priming was responsible for the values we saw. It has been shown for enteroviruses that false-priming, an event in which the viral RNA uses other RNA molecules as primers to synthesize cDNA during the reverse transcription step of RT-qPCR, can lead to false positive results and/or overestimation of the actual quantity of RNA molecules present in the sample [36]. False-priming is most probably the reason we saw the low amounts of −RNA at 0 h p.i. in our results, but since false-priming occurred in all of the samples, the results were still comparable with each other. In future studies, it is recommended that measures (such as tagged RT-qPCR, as suggested in [36]) be taken in order to avoid false-priming in RT-qPCR.

We demonstrate here the use of bDNA FISH in the study of enterovirus RNA strands during an infection and established a working protocol for the method. Even though we had some problems with the visibility of the −RNA, the use of 95% formamide in 0.1x SSC buffer improved the visibility enough that we were able to study the −RNA effectively. As mentioned, we also tested ice-cold 100% methanol and up to 8 molar urea, but these chemicals had no effect on the visibility of the −RNA. It has been reported that dimethyl sulfoxide (DMSO) is more effective than formamide in the separation of DNA strands [37], but unfortunately we did not have time to test DMSO in our study. Due to the fragility of RNA, physical separation methods such as beads mill or sonication will probably not work with this protocol, but DMSO could provide a safer and more effective visibility improving agent for bDNA FISH than formamide. More studies are needed to properly characterize the effects of DMSO on the visibility of enterovirus −RNA strands while using bDNA FISH. Importantly, the addition of incubating the cells in 95% formamide in 0.1x SSC-buffer at 65 °C to the protocol was the key aspect of improving the visibility of the −RNA, as was already suggested for poliovirus FISH in earlier literature [38].

Since the ViewRNA Cell Plus Assay Kit allows for the labelling of only three different molecules, we unfortunately could not label the capsid, +RNA, −RNA, and dsRNA all at once. However, our results suggest that the −RNA and dsRNA, which take part only in replication, are located quite centrally in the cell, indicating that the virus’ replication organelles (ROs) are located centrally in the cell as well. Along with capsid proteins, the +RNA is located peripherally (but also quite ubiquitously later in the infection) in the cell, indicating that the assembly of new virus particles takes place in the cell periphery. These findings are consistent with earlier literature describing ROs being usually located close to the perinuclear endoplasmic reticulum [39,40] and virion assembly taking place closer to the peripheral endoplasmic reticulum [40]. Since the dsRNA resides close to the −RNA, we suspect that most of the dsRNA strands we see are replication intermediates. Enterovirus RNA and/or protein localization studies were not performed with the paraffin-embedded samples at later time points. Enteroviruses do also produce stable replicative form dsRNA structures whose function is yet to be discovered [35].

We demonstrated that the antivirals Rac1-I and Cal-I1 effectively blocked the CVA9 infection in cells in vitro at 200 µM concentrations. It is noteworthy to mention that the images taken from the bDNA FISH experiment with the antivirals gave only a glimpse of the reality of the situation; even though the amount of RNAs in Figure 8 might seem high, only a small percentage of cells exhibited viral RNAs. The RT-qPCR experiment (Figure 7) gave a better idea of the effectiveness of the antivirals tested as a whole. We thus showed here that bDNA FISH and RT-qPCR are both suitable methods for studying the effects of antiviral drugs on cells infected with enteroviruses. Both drugs that we tested are promising in vitro, but require more studies to study their toxicity, their effectiveness at lower concentrations, and their possible delivery methods in vivo.

## 5. Conclusions

The results demonstrate that ViewRNA ISH with our custom probes is a suitable method to study the expression of +RNA and −RNA in acute and persistent enterovirus infection. The same probe sets detecting +RNA and −RNA of enteroviruses can be applied across ViewRNA platforms, and the method works well also for FFPE tissue samples. According to our results, the replication of the viral RNA starts 2–3 h after infection and the translation of RNA to capsid proteins starts shortly after at 3–4 h post-infection. The virus’ replication areas with newly emerging −RNA are located centrally in the cell, while the virion assembly most likely takes place in the periphery of the cell. We also discovered that the pace of replication of −RNA and +RNA strands is almost identical, contradictory to earlier literature. Considering these results, new antiviral drugs can be designed more efficiently to target the various steps of the enterovirus life cycle. Moreover, the bDNA FISH protocol we established can be easily and readily applied to study enterovirus RNAs in vitro more effectively.

## Figures and Tables

**Figure 1 microorganisms-08-01928-f001:**
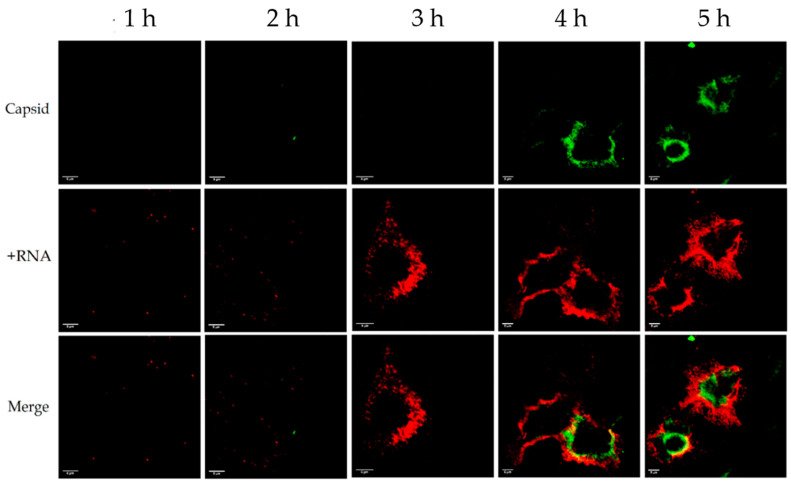
Distribution of viral +RNA and capsid protein in coxsackievirus A9 (CVA9)-infected cells 1–5 h post-infection (p.i.). The capsid protein is shown in green and the +RNA is shown in red. The rows from left to right are 1, 2, 3, 4, and 5 h p.i. Capsid protein became visible after 4 h p.i. and +RNA was visible at all timepoints, but surged in visibility at 3 h p.i. Around 30% of the capsid colocalized with the +RNA. Scale bar in every image is 5 µm.

**Figure 2 microorganisms-08-01928-f002:**
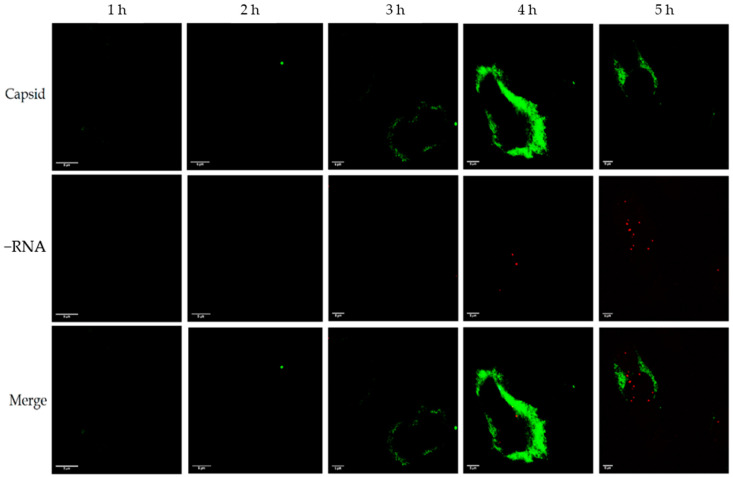
Distribution of viral −RNA and capsid protein in CVA9-infected cells 1–5 h p.i. The capsid protein is shown in green and the −RNA is shown in red. The rows from left to right are 1, 2, 3, 4, and 5 h p.i. Capsid protein became visible after 3–4 h p.i. and −RNA became visible at 4 h p.i. There was almost no colocalization between the capsid and the −RNA. Scale bar in every image is 5 µm.

**Figure 3 microorganisms-08-01928-f003:**
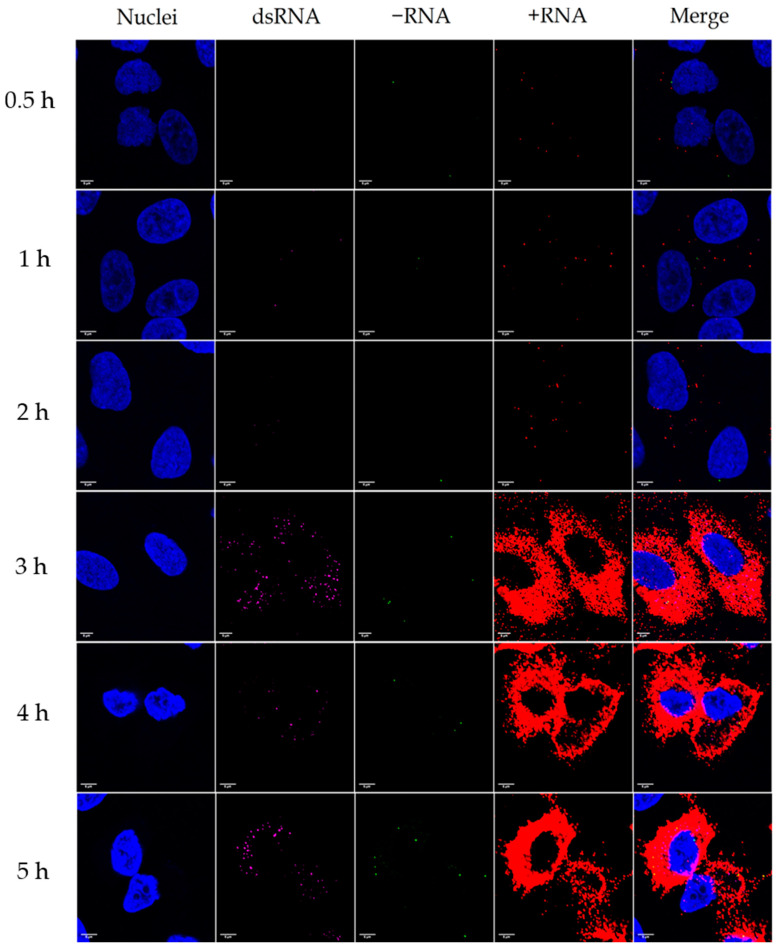
Distribution of viral −RNA, +RNA, and double-stranded RNA (dsRNA) in CVA9-infected cells 0.5–5 h p.i. Nuclei are shown in blue, dsRNA is shown in magenta, −RNA is shown in green, and +RNA is shown in red. The rows from top to bottom are 0.5, 1, 2, 3, 4, and 5 h p.i. Some dsRNA could be seen at 0.5–2 h p.i., but the amount of dsRNA surged at 3 h p.i. The amount of −RNA at timepoints 0.5–2 h p.i was negligible, but some −RNA could be seen at timepoints 3–5 h p.i. The +RNA from the input virus could be seen at timepoints 0.5–2 h p.i., and the amount of +RNA surged at 3 h p.i. and gradually increased at timepoints 4 h and 5 h p.i. Only about 1% of the −RNA colocalized with the dsRNA, while 97% of the −RNA colocalized with the +RNA, and 89% of the dsRNA colocalized with the +RNA. The images are typical views of 2–3 cells. Scale bar in every image is 5 µm.

**Figure 4 microorganisms-08-01928-f004:**
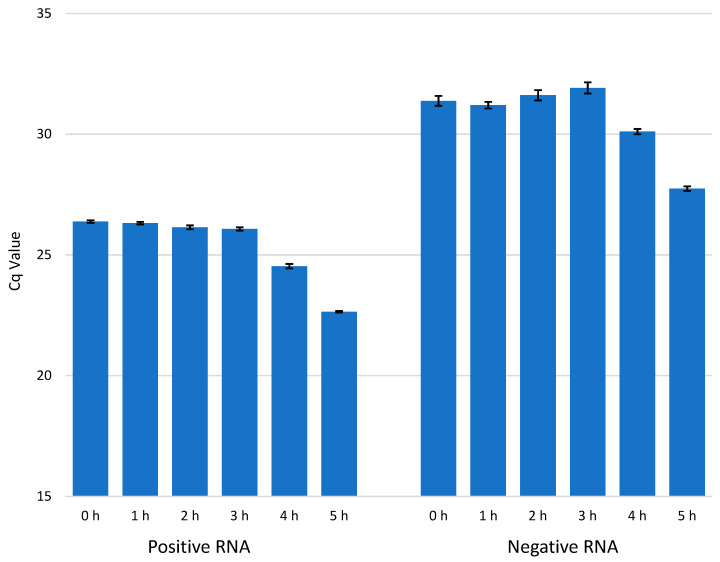
The amount of viral +RNA and −RNA in CVA9-infected cells at timepoints 0–5 h p.i. The amount of +RNA at timepoints 0–5 h p.i. is shown on the left, and the amount of −RNA at timepoints 0–5 h p.i. is shown on the right. The bars are averages calculated from triplicate samples, and the error bars are standard deviations from the average values.

**Figure 5 microorganisms-08-01928-f005:**
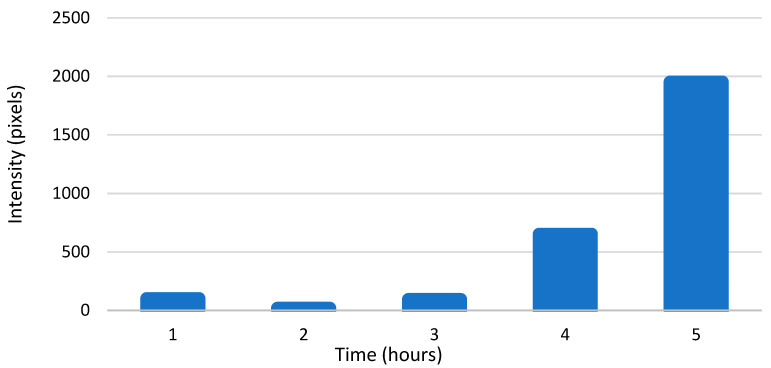
The amount of viral +RNA in CVA9-infected cells at timepoints 0–5 h p.i., as calculated from branched DNA (bDNA) fluorescent in situ hybridization (FISH) image intensity values. The bars represent pixels that have intensity values above the background threshold, calculated from ≈100 cells.

**Figure 6 microorganisms-08-01928-f006:**
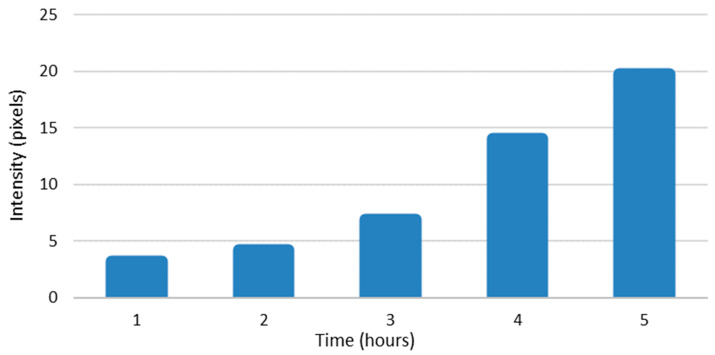
The amount of viral −RNA in CVA9-infected cells at timepoints 0–5 h p.i., as calculated from bDNA FISH image intensity values. The bars represent pixels that have intensity values above the background threshold, calculated from ≈100 cells.

**Figure 7 microorganisms-08-01928-f007:**
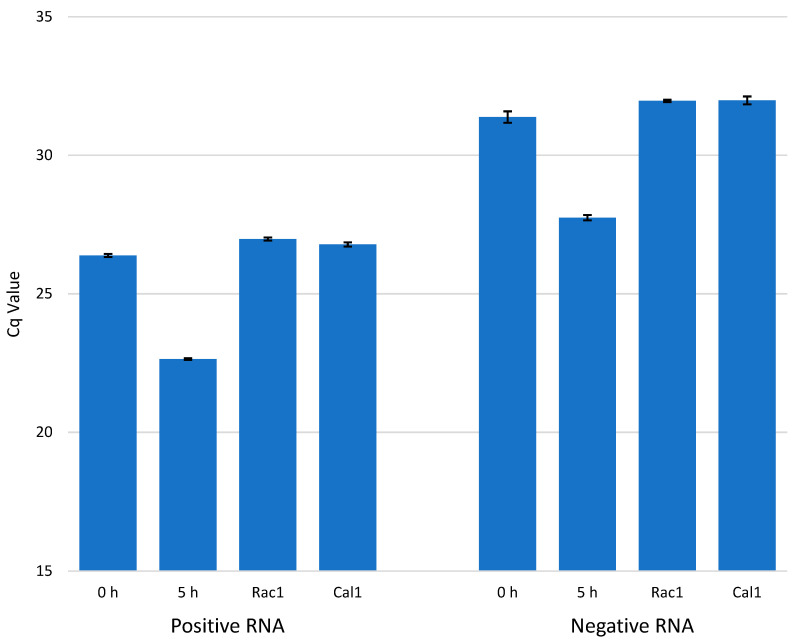
The effect of antiviral drugs Rac1 inhibitor NSC23766 (Rac1-I) and calpain inhibitor 1 (Cal-I1) on the amounts of enterovirus +RNA and −RNA in CVA9-infected cells at 5 h p.i. The amount of +RNA is shown on the left (samples from left to right: 0 h p.i. with no drugs, 5 h p.i. with no drugs, 5 h p.i with Rac1-I, 5 h p.i. with Cal-I1), and the amount of −RNA is shown on the right with the samples in the same order. The bars are averages calculated from triplicate samples, and the error bars are standard deviations from the average values.

**Figure 8 microorganisms-08-01928-f008:**
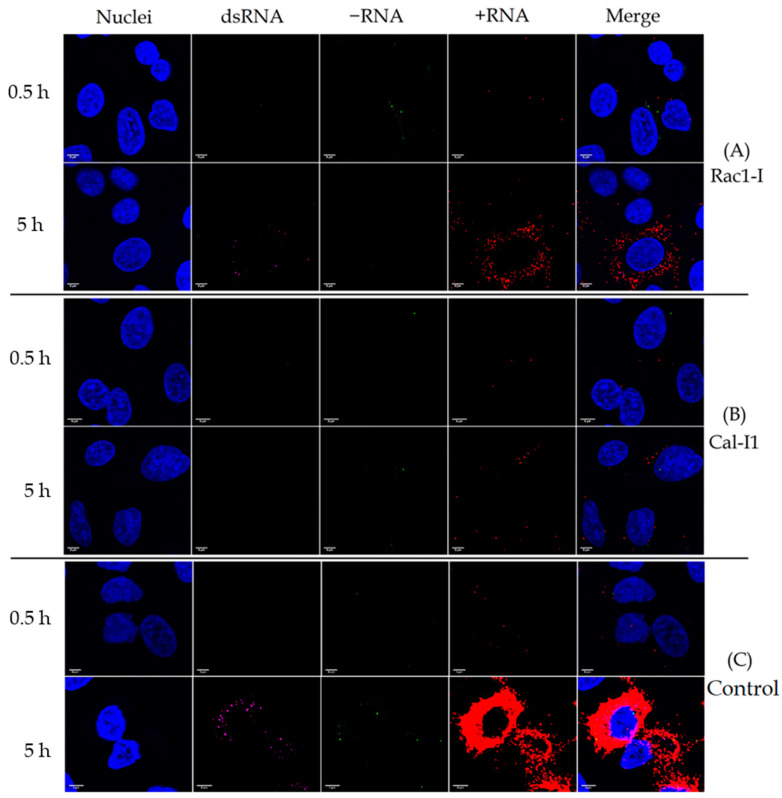
Effect of antiviral drugs Rac1-I and Cal-I1 and on the distribution on viral dsRNA, −RNA, and +RNA in CVA9-infected cells at timepoints of 0.5 h and 5 h p.i. Nuclei are shown in blue, dsRNA in magenta, −RNA in green, and +RNA in red. (**A**) Rac1-I inhibited the virus infection completely in most cells, but some cells exhibited input-virus levels of +RNA. (**B**) Cal-I1 inhibited the virus infection completely in almost all cells. (**C**) Control infection without the use of drugs showed the amount of viral dsRNA, −RNA, and +RNA in infected cells.

**Figure 9 microorganisms-08-01928-f009:**
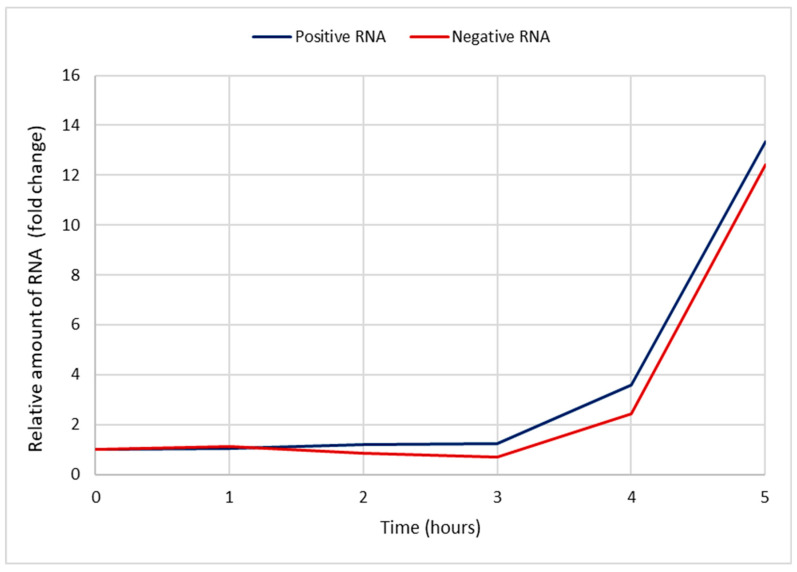
The increase in the relative amounts of viral +RNA and −RNA during a CVA9 infection compared to timepoint 0 h p.i. The data were plotted from the RT-qPCR results presented in Figure 4 by calculating the fold change with Equation (3). The *x*-axis shows the time post-infection, and the *y*-axis shows fold change (how many times higher the amount of RNA is compared to 0 h p.i.). The +RNA is shown in blue, and −RNA is shown in red. The pace at which the amounts of RNAs increased was almost identical.

**Figure 10 microorganisms-08-01928-f010:**
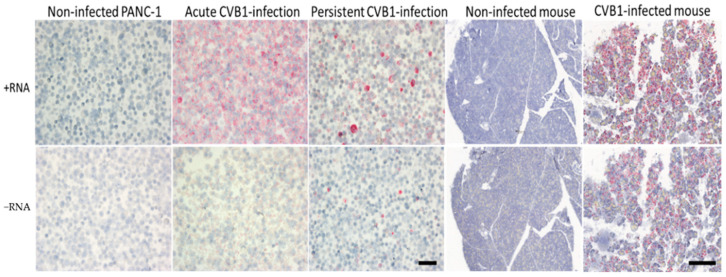
Acutely (1 d.p.i.) and persistently (172 d.p.i.) CVB1-infected pancreatic ductal (PANC-1) cells and acutely (3 d.p.i) CVB1-infected mouse pancreas with respective non-infected controls were paraffin embedded and stained for +RNA and −RNA using ViewRNA ISH. In the acute infections (cells and mice), both +RNA and −RNA were clearly detected (columns 2 and 5, respectively); however, the expression of +RNA was visibly stronger. Similarly, in persistent infection (column 3), both RNA strands were detected, and again +RNA was more abundant compared to −RNA. Representative images of non-infected cells and mouse pancreas are also shown (columns 1 and 4). These were negative for both RNA strands, as expected. Scale bar 50 µm in cell samples and 300 µm in mouse samples.

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
