# Peer review of "Detection of Viral −RNA and +RNA Strands in Enterovirus-Infected Cells and Tissues"

_microorganisms, 2020, doi:10.3390/microorganisms8121928_

Round 1

Reviewer 1 Report

Salmikangas et al. present an interesting manuscript analyzing the appearance of – and + RNA during acute and persistent enterovirus infections using branched DNA ISH technology (bDNA ISH). Interestingly – and + RNA we located at different sites within the cells and occurred in significantly different abundancies. Despites these differences the replication of -RNA and +RNA strands was almost identical. Viral replication could effectively blocked by Rac1 and calpain inhibitors.

The manuscript is well written, the instruction gives a good overview on the topics, the methods described in detail, the results clearly presented and the current literature thoroughly discussed,

The major questions arising is, whether the observed effects are strain specific? Have the authors observed differences used different strains?

Additionally, are the effects host cell specific? Have the authors tested differences?

Have the authors also checked later time-points than 5 hours? What happens with the -RNA?

How do the authors explain the different location of RNA and +RNA strands in their time-course experiments?

In Figure 8 the authors need to include the positive control, that should have been used at the same time to demonstrate that the experiment has worked.

In Figure 10 some aspects are unclear. Please clarify the legend. What is the difference between acute and persistent infections? What areas are shown in all images? I assume column 1-3 PANC-1 cells and column 4-5 the whole organ.

Author Response

Dear reviewer,

Thank you for your kind and insightful review report. Please see the response below where we have addressed your questions and revision suggestions in depth. We hope that you find these revisions satisfactory.

Point 1: The major questions arising is, whether the observed effects are strain specific? Have the authors observed differences used different strains?

Response: We did initial bDNA FISH experiments with Coxsackie virus A9 (CVA9), Coxsackie virus B3 (CVB3) and Echovirus 1 (EV1), and similar results were achieved with all virus strains. Due to the amount of work needed to do experiments with three different strains, we decided to carry out the rest of the experiments with CVA9 only. Mentions of the other strains used have been added on lines 295 and 307.

Point 2: Additionally, are the effects host cell specific? Have the authors tested differences?

Response: We have not tested other cell lines than A549 in this manuscript, but we expect that the results would be similar with other cell lines as well. Our antiviral targets Rac and Cal are expressed in all normal cell lines quite similarly, so we expect the antiviral drugs to perform similarly in other cell lines as well.

Point 3: Have the authors also checked later time-points than 5 hours? What happens with the -RNA?

Response: It would be interesting to look at later timepoints than 5 hours, but here we wanted to study the timepoints relevant for the start of replication and the antiviral effects of the drugs.

Point 4: How do the authors explain the different location of RNA and +RNA strands in their time-course experiments?

Response: These differences have been explained in the lines 512-517, and also linked to earlier literature (“These findings are consistent with earlier literature describing ROs being usually located close to the perinuclear endoplasmic reticulum [39-40] and virion assembly taking place closer to the peripheral endoplasmic reticulum [40]. Since the dsRNA resides close to the -RNA, we suspect that most of the dsRNA strands we see are replication intermediates. Enterovirus RNA and/or protein localization studies were not performed with the paraffin embedded samples at later time points. Enteroviruses do also produce stable replicative form dsRNA structures whose function is yet to be discovered [35]”).

Point 5: In Figure 8 the authors need to include the positive control, that should have been used at the same time to demonstrate that the experiment has worked.

Response: This is a very good point, and we have now included a positive control in the Figure 8. The positive control is the same images as the ones in figure 3 (timepoints 0.5h and 5h), since these experiments were done at the same time.

Point 6: In Figure 10 some aspects are unclear. Please clarify the legend. What is the difference between acute and persistent infections? What areas are shown in all images? I assume column 1-3 PANC-1 cells and column 4-5 the whole organ.

Response: The legend on figure 10 (lines 435-442) has been clarified. Days post infection (d.p.i) numbers have been added, and the different columns in the figure have been mentioned in the text for added clarity.

Reviewer 2 Report

In my opinion the paper is well written, descriptive and has a high scientific soundness. I would only like the Authors to work on the conclusions - in my opinion it can be modified to sound more firm and specific. The paper can be accepted.

Author Response

Dear reviewer,

Thank you for your kind review report. We have modified the conclusions of the manuscript to sound more firm and the text to flow more easily.

Reviewer 3 Report

The authors submit a manuscript that uses semi-quantitative RT-PCR and bDNA FISH to identify - and + stranded viral RNA in cells grown in culture and in mouse tissues. Although the findings are provocative there are several criticisms that would have to be addressed prior to publication.

  1. Introduction
    1. The introduction is well written and the hypothesis is clear
  2. Materials and Methods
    1. There is not a sufficient description of how the authors were able to differentiate dsRNA from - and + stranded RNA. What probes were used to differentiate the 3 types? A list of probes is given for - and + stranded RNA but it is not clear what probe was used to obtain the magenta color in Figure 3.  
  3. Results
    1. The images are well presented and are clear, however it would be helpful to include the DAPI/nuclear stain in Figure 1 and 2 to help the reader understand subcellular localization and to be able to differentiate cell membrane, cytoplasm and nuclei.
    2. The authors make statements on lines 293-295 regarding the localization of RNA's. This conclusion is drawn based on very few cells presented. They state that approximately 100 cells per image were analyzed on line 255. A quantitative assessment of these percentages of -RNA and +RNA that colocalize with VP1 would validate this point.
    3. In Figure 3 it would be helpful to understand the signal overlap between dsRNA, -RNA and + RNA. In line 307 the authors state that "the dsRNA resides mostly in the same location as the -RNA". This should be quantitated more formally.
    4. In Figure 3 it is unclear from the figure legend , results, materials and methods and/or appendix how they authors were able to distinguish dsRNA from -RNA or +RNA. This is a key part of this manuscript, so further clarification would be helpful.
    5. In Figure 4 I would like to see an internal control RNA amplification or a housekeeping gene amplification to verify the quality of the RNA and further give validity to the quantitative aspects of this manuscript. Alternatively, the authors could use a viral RNA standard derived from RNA amplified off of a plasmid.
    6. Figure 5 and 6. Although the bDNA FISH allows the detection of very small amounts of RNA and is very sensitive it is not clear whether the amplification of the signal is a linear function. In that case quantitation of the signal would not be valid.
    7. Figure 7. These anti-viral drugs have several downstream effects on cells in culture and it is unclear whether the drugs RacI and CalI act directly on virus or whether this effect is due to cell toxicity. The use of a housekeeping gene in the qRT-PCR or performing a cell viability assay after 5 hours would further convince this reviewer that these drugs are behaving as anti-virals in this assay. In addition a dose titration experiment would improve this.
    8. Figure 8. Very few cells presented here. A quantitative assessment would improve this.

Author Response

Dear reviewer,

Thank you for your kind and insightful review report. Please see the response below where we have addressed your questions and revision suggestions in depth. We hope that you find these revisions satisfactory.

Point 1: There is not a sufficient description of how the authors were able to differentiate dsRNA from - and + stranded RNA. What probes were used to differentiate the 3 types? A list of probes is given for - and + stranded RNA but it is not clear what probe was used to obtain the magenta color in Figure 3.

Response: The dsRNA was detected with immunofluorescence with the primary antibody J-2 mentioned in the supplementary material. This has now been indicated more clearly on lines 180 and 314.

Point 2: The images are well presented and are clear, however it would be helpful to include the DAPI/nuclear stain in Figure 1 and 2 to help the reader understand subcellular localization and to be able to differentiate cell membrane, cytoplasm and nuclei.

Response: Unfortunately, we could not include a nuclear stain in figures 1 and 2 due to channel limitations on the microscope used. The subcellular localizations of all of the RNA molecules can be better seen in figure 3.

Point 3: The authors make statements on lines 293-295 regarding the localization of RNA's. This conclusion is drawn based on very few cells presented. They state that approximately 100 cells per image were analyzed on line 255. A quantitative assessment of these percentages of -RNA and +RNA that colocalize with VP1 would validate this point.

Response: Colocalizations were quantitated with Manders’ Colocalization Coefficients, and results were added on lines 294 and 306. Information about performing the colocalization analyses were added on lines 257-259.

Point 4: In Figure 3 it would be helpful to understand the signal overlap between dsRNA, -RNA and + RNA. In line 307 the authors state that "the dsRNA resides mostly in the same location as the -RNA". This should be quantitated more formally.

Response: Colocalizations were quantitated here as well, and results were added on lines 319-326.

Point 5: In Figure 3 it is unclear from the figure legend, results, materials and methods and/or appendix how they authors were able to distinguish dsRNA from -RNA or +RNA. This is a key part of this manuscript, so further clarification would be helpful.

Response: The dsRNA was detected with the primary antibody J-2. This has now been indicated more clearly on lines 178-180 and 314.

Point 6: In Figure 4 I would like to see an internal control RNA amplification or a housekeeping gene amplification to verify the quality of the RNA and further give validity to the quantitative aspects of this manuscript. Alternatively, the authors could use a viral RNA standard derived from RNA amplified off of a plasmid.

Response: We have now performed a standard curve from RNA purified from CVA9. This helps to evaluate the RNA levels of CVA9 detected in cells after various time periods. The standard curve has been added in the Supplementary figure 2, and mentioned in the manuscript on the lines 345-348.

Point 7: Figure 5 and 6. Although the bDNA FISH allows the detection of very small amounts of RNA and is very sensitive it is not clear whether the amplification of the signal is a linear function. In that case quantitation of the signal would not be valid.

Response: According to the manufacturer, each dot in bDNA FISH images represents one target transcript, and the level of expression can be quantified by measuring the number of dots per cell, meaning that the amplification of the signal is a linear function. This has now been added to the lines 264-268. However, it is important to note that the experimental conditions, such as the microscope and camera used, can affect the visibility of the RNAs, meaning that the results are comparable only with studies with similar settings.

Point 8: Figure 7. These anti-viral drugs have several downstream effects on cells in culture and it is unclear whether the drugs RacI and CalI act directly on virus or whether this effect is due to cell toxicity. The use of a housekeeping gene in the qRT-PCR or performing a cell viability assay after 5 hours would further convince this reviewer that these drugs are behaving as anti-virals in this assay. In addition, a dose titration experiment would improve this.

Response: We did a cell viability assay to determine whether the drugs Rac1-I and Cal-I1 act directly on the virus, or if their effect is due to cytotoxicity. After a 5-hour incubation with the drugs, only a 10% drop in cell viability was detected with Cal-I1, and no noticeable drop in cell viability was detected with Rac1-I, compared to a DMSO control (since the drugs are dissolved in DMSO). This proves that rather than having cytotoxic effects, the drugs act directly as antivirals. These results have been added to lines 374-378, and a supplementary figure of the assay results has been provided (Supplementary figure 1).

Point 9: Figure 8. Very few cells presented here. A quantitative assessment would improve this.

Response: Very few cells have been presented since the -RNA signal is very weak in the drug treated cells and can only be seen on high magnifications. In our opinion, qPCR is better fit for quantitative assessments than bDNA FISH image intensity analysis. The qPCR quantitation of the amounts of RNA molecules in drug treated cells is presented in figure 7.

Round 2

Reviewer 1 Report

The reviewer comments were adressed.

Reviewer 3 Report

The authors have sufficiently addressed my prior concerns. These additions and corrections significantly improve this work.